# `tsGT`: Time Series Generative Transformer

## Abstract

Time series models are ubiquitous in fields of science that deal with temporally structured data. Recent advancements in time series analysis have seen a growing trend toward the popularity of tailor-made transformer neural networks with customized attention blocks and hand-crafted intricate design features. We show, perhaps surprisingly and against this current trend, that a simple time series generative transformer model, dubbed `tsGT`, based on a vanilla decoder-only architecture with the discretization of real values outperforms more sophisticated contemporary models on selected prediction tasks. We evaluate `tsGT` against eleven baselines and show that it surpasses its deterministic peers on MAE and RMSE, and the stochastic ones on QL and CRPS on four commonly used datasets: `electricity`, `traffic`, `ETTm2`, and `weather`. We use a well-known and theoretically justified rolling window evaluation protocol and provide a detailed analysis of `tsGT`'s ability to model the data distribution and predict marginal quantile values. We provide an implementation of our method at `https://github.com/ts-gt/tsgt`.

## 1 Introduction

Researchers and practitioners use data in the form of time series to predict and analyze future events, estimate relationships, and make informed decisions across a diverse range of scientific fields. With applications spanning medicine, finance, economics, meteorology, astronomy, transportation, and manufacturing, the significance of time series methods cannot be overstated.

The generative mechanism behind time series data is often modeled as an autoregressive stochastic process (Hamilton, 2020). These models encompass a broad range of approaches, such as classical econometrics techniques like ARMA (Whittle, 1951) and ARCH models (Engle, 1982), or recurrent neural networks (Hochreiter & Schmidhuber, 1997) and transformers (Vaswani et al., 2017) in deep learning. In particular, the last class of models emerged as a powerful architecture that fuels many of the recent advances in machine learning, including text generation (Brown et al., 2020), solving mathematical problems (Lewkowycz et al., 2022), hyperparameter tuning (Chen et al., 2022), or EEG analysis (Kostas et al., 2021).

Transformers have also made their way to the time series domain (Li et al., 2019; Woo et al., 2022; Zhou et al., 2022; Wu et al., 2021; Zhou et al., 2021; Wen et al., 2023; Drouin et al., 2022; Ashok et al., 2024; Rasul et al., 2024). Interestingly, contemporary models often feature domain-specific design choices in neural network architectures, input processing, or loss function formulation. Many models are deterministic, and the evaluation protocol is often limited to using pointwise metrics within a single time window. Some of these choices offer benefits, including simplicity, ease of interpretation, and computation efficiency. However, in this paper, we focus on a general approach to time series modeling and a well-established evaluation protocol.

We present a simple time series generative transformer, dubbed `tsGT`, based on a vanilla decoder-only transformer with a real value discretizer. `tsGT` is stochastic and uses a general-purpose architecture without any domain-specific complex inductive biases. This approach has several important benefits, including the reduced cost of training due to the decreased number of dataset-specific hyperparameters and facilitates effective scaling with data, computation, and model size (Sutton, 2019; Kaplan et al., 2020; Hoffmann et al., 2022). It also allows us to incorporate any progress in transformer efficiency, which is an active field of research (Tay et al., 2022; Zhou et al., 2024; Tang et al., 2024), directly into `tsGT`.

We show that the simplicity of `tsGT` leads to a considerable increase in performance on selected prediction tasks in four commonly used datasets: `electricity`, `traffic`, `ETTm2`, and `weather`, with respect to eleven baseline models. `tsGT` significantly outperforms its deterministic peers on mean absolute error (MAE) and root means square error (RMSE) metrics and its stochastic peers on quantile loss (QL) and continuous ranked probability score (CRPS). Importantly, we focus on testing the models' predictive capabilities, employing a rolling window analysis, which is a well-known time-series procedure that is robust to outliers and assesses the methods' stability over time. In particular, this allows us to backtest the stochastic modeling abilities of `tsGT` on a granular, per-quantile level (McNeil et al., 2015) and demonstrate that for higher quantile values `tsGT` outperforms its stochastic peers. Finally, we demonstrate that `tsGT` preserves temporal information and does not adhere to a recent critique of transformers models for time series modeling Zeng et al. (2023).

Our contributions are as follows:

- We propose `tsGT`, a general-purpose transformer stochastic time series model. Additionally, we focus on using a rolling window evaluation protocol and backtesting the predictive performance.

- We show that in terms of MAE and RMSE, `tsGT` outperforms contemporary deterministic transformer models. Interestingly, `tsGT` uses the same training hyperparameters across the datasets.

- We rigorously assess `tsGT`'s predictive performance. In particular, we show that `tsGT` outperforms stochastic baselines on QL and CRPS, analyze `tsGT`'s ability to model data distribution, and demonstrate that it preserves temporal information.

## 2 Related work

The classical treatment of time series modeling can be found in (Hamilton, 2020; Brockwell & Davis, 2009; Hyndman & Athanasopoulos, 2021). Historically, much of the progress in the field has been driven by financial and risk management applications (Nobel Foundation, 2003; Zivot & Wang, 2006; Tsay, 2010; McNeil et al., 2015). More recently, deep learning methods (Goodfellow et al., 2016), such as RNNs and Transformers have made their way into the field (Salinas et al., 2020; Li et al., 2019). Transformers are strong sequence-to-sequence architectures, and their large-scale variants are known as large language models (LLMs). There is an overwhelming abundance of LLMs, including GPT variants (Radford et al., 2019; Brown et al., 2020; OpenAI, 2023), PaLM (Chowdhery et al., 2022), LLaMA (Touvron et al., 2023), or Gemini (Team et al., 2023). See (Liang et al., 2022; Zhao et al., 2023; Yang et al., 2023) for recent surveys on the topic. Adaptation of LLMs to time series tasks has been explored in (Gruver et al., 2024; Wu et al., 2023). While our work focuses on training transformer models, which are smaller then modern LLMs by several orders of magnitude, we employ architectural choices found to be useful in LLMs, including rotary embeddings (Su et al., 2021) and GELU activation function (Hendrycks & Gimpel, 2016).

Following the popularity of transformers, there has been a surge of transformer-inspired time series models; see (Wen et al., 2023) for a recent survey. Recent deterministic models include ETSformer (Woo et al., 2022), FEDformer (Zhou et al., 2022), Autoformer (Wu et al., 2021), Informer (Zhou et al., 2021), PatchTST (Nie et al., 2022), Crossformer (Zhang & Yan, 2023). Some of these models were criticized by Zeng et al. (2023), as they were shown to be permutationally invariant in some evaluation settings and to fall short of a simple MLP-based model (DLinear). Other all-MLP models for time series include N-BEATS (Oreshkin et al., 2020), N-HiTS (Challu et al., 2022) and TSMixer (Chen et al., 2023). Stochastic transformer time series models include ConvTrans (Li et al., 2019), Temporal Fusion Transformer (Lim et al., 2021), IQN-Transformer (Gouttes et al., 2021), FQFormer (Jawed & Schmidt-Thieme, 2022), Tactis (Drouin et al., 2022; Ashok et al., 2024). Multivariate modeling of time series has been explored, and joint distribution of variates has been approached via Gaussian Processes (Salinas et al., 2019), attention copulas, hypernetworks, deep sigmoidal flows (Drouin et al., 2022; Ashok et al., 2024), diffusion models (Rasul et al., 2021a), GANs (Brophy et al., 2021), and normalizing flows (Rasul et al., 2021b).

Recently, first time series foundation models trained on vast datasets started to emerge: Lag-Llama (Rasul et al., 2024), which also uses decoder-only architecture but does not discretize real values, contains parametric

distribution heads, and heavily relies on cyclic, lag, and summary statistics covariates. Other attempts to create time series foundational models Garza & Mergenthaler-Canseco (2023); Das et al. (2023).

There are multiple ways of handling real-valued data in the literature. Typically, the raw time series values are normalized before further processing, e.g., by scaling to a certain interval (Li et al., 2019) or centered and standardized (Nie et al., 2022). After that, the data is tokenized, via bucketing (Chen et al., 2021; Janner et al., 2021; Chen et al., 2022; Reed et al., 2022) or $k$-means (Shafiullah et al., 2022), or directly embedded as a multivariate vector e.g., by introducing covariates (Li et al., 2019), modeling multivariate time series (Zhou et al., 2021; Wu et al., 2021; Zhou et al., 2022; Woo et al., 2022), or patching (Nie et al., 2022; Zhou et al., 2023). `tsGT` handles preprocessing of real-valued data in three steps: (a) the values are normalized to $[0, 1]$ interval, similar to (Li et al., 2019), (b) the values are quantized into a fixed precision, and (c) the values are split into digits. The last step is similar to (Nogueira et al., 2021; Chowdhery et al., 2022; Gruver et al., 2024), where tokenization of string representation of numbers into digits was used to improve parsing simple text-based math problems. Steps (b) and (c) can also be viewed as a form of vector quantization (VQ) (Gray, 1984; Lendasse et al., 2005; Oord et al., 2017), although VQ is often used differently, i.e., to compress the data into a finite set of prototype vectors; see also VQ-AR (Rasul et al., 2022) in the context of time series modeling.

The contemporary time series methods mentioned above are trained and evaluated on a fixed time window. In this paper, we perform a rolling-window analysis, i.e., we train and evaluate models on multiple overlapping windows (McNeil & Frey, 2000; González-Rivera et al., 2004; McNeil et al., 2015). This allows assessing the methods' stability over time and their predictive performance via backtesting procedure (McNeil et al., 2015). Finally, we leave the efficient transformer implementation out of the paper's scope, since it is a challenge studied intensively across the deep learning field, see e.g., (Tay et al., 2022). This allows us to focus on modeling complex stochastic properties of continuous data using a plug-and-play design, which facilitates the integration of efficiency-related innovation in the field.

## 3   Time series model: `tsGT`

**Time series notation**   We consider a time series which ends at time $t$ and has a length $T$, $\mathbb{X}_{t-T+1:t} = (X_{t-T+1}, \ldots, X_t)$. Our goal is to predict the $H$ timesteps following $t$, i.e., $\mathbb{X}_{t+1:t+H} = (X_{t+1}, \ldots, X_{t+H})$. The raw data $\mathbb{X}_{t-T+1:t}$ forms an input to the model, and no additional covariates are provided (such as series id, timestamp, or other exogenous factors).

**Transformer backbone**   Motivated by strong sequence-to-sequence Transformers performance, we use a decoder-only transformer (Vaswani et al., 2017), with pre-normalization (Radford et al., 2019), rotary embeddings (Su et al., 2021), and GELU activation function (Hendrycks & Gimpel, 2016). These choices are inspired by recent successful architectures like GPT-3 (Brown et al., 2020) or LLaMA (Touvron et al., 2023). Notice that we do not use time series-specific architectural modifications, such as 1D convolutions (Li et al., 2019), frequency domain (Wu et al., 2021; Zhou et al., 2022), seasonal-trend decomposition (Woo et al., 2022), patching (Nie et al., 2022), copulas, hypernetworks, or Deep Sigmoidal Flows (Drouin et al., 2022; Ashok et al., 2024). For architectural details, see Appendix A-B.

**Input and output discretization**   Following the success of transformers in language modeling, where the input consists of sequences from a finite alphabet, we discretize the real-valued data into a stream digits. Discretization is performed in three stages: (a) normalization of data, (b) quantization into fixed precision, and (c) discretization into digits.

During the stages (a) and (b), the values are divided by a normalizing constant and squeezed to the interval $[0, 1]$, a procedure also found in (Salinas et al., 2020; Janner et al., 2021; Chen et al., 2022; Reed et al., 2022), see Appendix A.

Next, each value $a \in [0, 1]$ is tokenized to digits in a fixed precision $p \in \mathbb{N}_+$ in base $B > 1$, via a function $\tau(a) = (\tau_1(a), \ldots, \tau_p(a))$, defined through an equation $\lfloor a \cdot B^p \rfloor = \sum_{k=1}^{p} \tau_k(a) \cdot B^{p-k}$. For example, with $p = 3$ and $B = 10$, a number $a = 0.123$ is represented as three tokens 1, 2, 3.

With a slight abuse of notation, we denote by $\tau(\mathbb{X})$ a series of tokens resulting from breaking a normalized time series $\mathbb{X}$ into digits (with a length of $\tau(\mathbb{X})$ being $p$ times larger than the length of $\mathbb{X}$). For each $k$, a model $g_\theta$ outputs a conditional distribution $g_\theta(\cdot|\tau_1, \ldots, \tau_k)$ over the next $(k+1)$-th token (digit). If we denote $\tau' = (\tau'_1, \ldots, \tau'_{pT+pH}) = \tau(\mathbb{X}_{t-T+1:t+H})$, the conditional distribution of the predicted horizon is given by

$$g_\theta(\tau'_{pT+1}, \ldots, \tau'_{pT+pH}|\tau'_1, \ldots, \tau'_{pT}) = \prod_{k=pt+1}^{pT+pH} g_\theta(\tau'_k|\tau'_1, \ldots, \tau'_{k-1}).$$

The distribution of each real-valued number returned by the model is induced by the joint probability of $p$ consecutive digits and the mapping inversing the tokenization procedure. This joint probablity is recovered via a chain rule as a product of non-parametric conditional distributions returned by the model. This procedure gives the model probabilistic flexibility, allowing a range of complex and multi-modal distributions.

**Loss function**  We use a standard transformer training objective of predicting the next token with a weighted cross-entropy loss:

$$\mathcal{L}(\theta) = \mathbb{E}_{\substack{\mathbb{X} \sim Data \\ \tau' = \tau(\mathbb{X})}} \left[ \sum_{k=1}^{pT} \beta^{(k-1)\%p} \log g_\theta(\tau'_k|\tau'_1, \ldots, \tau'_{k-1}) \right]. \tag{1}$$

Here $\beta \in (0, 1]$ is a hyperparameter discriminating between the significance of digits, and $k\%p$ denotes the remainder of $t$ divided by $p$.

**Simulation**  For clarity of exposition, we omit discretization details in what follows, letting the reader infer specific details. The predictions are generated autoregressively using a standard Monte Carlo method. Assuming that for a time series of length $T$ ending at time $t$, $\mathbb{X}_{t-T+1:t}$, we have already generated $h$ steps of predictions, we generate the next value, $X_{t+h+1}$, as the sample from the model evaluated on $\mathbb{X}_{t-T+1:t+h}$. Consequently, we append $X_{t+h+1}$ to the sequence, resulting in $\mathbb{X}_{t-T+1:t+h+1}$, and the procedure is repeated. In this paper, we make $I = 1024$ simulations of $H$ steps following $t$, for every considered $t$.

## 4 Experimental setup

Below, we present the setup used for our experiments, postponing experimental results to the next section. In Section 5, our main result shows the `tsGT` outperforms the baselines on all considered datasets and metrics.

**Transformer architecture**  We set the precision $p = 3$ and base $B = 10$. During training, the raw time series input window is $T = 256$, which translates to $pT = 768$ tokens fed to the transformer. During inference, we set the prediction horizon $H = 24$, and the raw time series input window equals $T = 256 - H = 232$. In all our experiments, the transformer decoder comprises 6 layers and 4 self-attention heads, with an embedding dimension of 256. Our models have $\approx 3.2$M trainable weights. For training, we use a batch size 16, Adam optimizer, and scheduling the learning rate (multifactor approach with a constant of 0.03, linear warmup of 1000 steps, and square root decay). Additionally, we apply a weight decay with the rate of $10^{-5}$ and do not use early stopping. See Appendix B for more details.

**Baseline models**  We include as baselines several contemporary transformer and non-transformer methods. Models from the former category include FEDformer (Zhou et al., 2022), Informer (Zhou et al., 2021), Autoformer (Wu et al., 2021), ETSformer (Woo et al., 2022), IQN-Transformer (Gouttes et al., 2021), PatchTST (Nie et al., 2022), and a family of stochastic models inspired by Li et al. (2019). This family includes variants with different parametric marginal distributions: Gaussian, Laplace, and t-student. Non-transformer models include DLinear (Zeng et al., 2023) and TSMixer (Chen et al., 2023), representing linear and MLP-based approaches, respectively. Among the baselines, IQN-Transformer, Gaussian, Laplace, and t-student are stochastic, and other ones are deterministic. For the deterministic models, we use hyperparameters proposed by their open-source implementations. For the details on the hyperparameters used, see Appendix B. The models' approximate parameter count is as follows: FEDformer 18M, Informer 12M, Autoformer 11M, and ETSformer 6M. For more details on baselines, see Appendix C-D, and Appendix G for computational infrastructure used.

Table 1: Performance of models in terms of error metrics MAE and RMSE (lower values are better). Each entry is computed using IQM and accompanied by a 90% bootstrap confidence interval (in brackets) computed over $W = 100$ training windows and all time series.

| metric | model | electricity | traffic | ETTm2 | weather |
|--------|-------|-------------|---------|-------|---------|
| MAE | t-student* | $0.126_{(0.125,\ 0.126)}$ | $0.273_{(0.272,\ 0.274)}$ | $0.187_{(0.172,\ 0.204)}$ | $0.125_{(0.116,\ 0.134)}$ |
| | Laplace* | $0.127_{(0.126,\ 0.127)}$ | $0.272_{(0.272,\ 0.273)}$ | $0.186_{(0.171,\ 0.204)}$ | $0.125_{(0.116,\ 0.134)}$ |
| | Gaussian* | $0.125_{(0.124,\ 0.125)}$ | $0.275_{(0.274,\ 0.276)}$ | $0.182_{(0.168,\ 0.199)}$ | $0.135_{(0.126,\ 0.145)}$ |
| | IQN* | $0.124_{(0.123,\ 0.124)}$ | $0.349_{(0.348,\ 0.351)}$ | $0.183_{(0.169,\ 0.200)}$ | $0.129_{(0.121,\ 0.137)}$ |
| | Informer | $0.235_{(0.233,\ 0.236)}$ | $0.195_{(0.193,\ 0.196)}$ | $0.195_{(0.194,\ 0.197)}$ | $0.224_{(0.223,\ 0.226)}$ |
| | ETSformer | $0.201_{(0.200,\ 0.203)}$ | $0.192_{(0.191,\ 0.193)}$ | $0.193_{(0.192,\ 0.195)}$ | $0.206_{(0.204,\ 0.207)}$ |
| | FEDformer | $0.179_{(0.178,\ 0.180)}$ | $0.162_{(0.161,\ 0.164)}$ | $0.156_{(0.154,\ 0.157)}$ | $0.180_{(0.178,\ 0.181)}$ |
| | Autoformer | $0.169_{(0.167,\ 0.170)}$ | $0.161_{(0.159,\ 0.162)}$ | $0.152_{(0.151,\ 0.154)}$ | $0.173_{(0.172,\ 0.174)}$ |
| | DLinear | $0.065_{(0.064,\ 0.065)}$ | $0.181_{(0.180,\ 0.182)}$ | $\mathbf{0.094}_{(0.089,\ 0.101)}$ | $0.111_{(0.105,\ 0.118)}$ |
| | PatchTST | $0.062_{(0.062,\ 0.062)}$ | $0.152_{(0.152,\ 0.153)}$ | $\mathbf{0.094}_{(0.089,\ 0.101)}$ | $0.079_{(0.073,\ 0.085)}$ |
| | TSMixer | $0.063_{(0.062,\ 0.063)}$ | $0.153_{(0.152,\ 0.154)}$ | $\mathbf{0.090}_{(0.084,\ 0.096)}$ | $0.087_{(0.081,\ 0.092)}$ |
| | tsGT* | $\mathbf{0.057}_{(0.057,\ 0.058)}$ | $\mathbf{0.114}_{(0.114,\ 0.115)}$ | $0.092_{(0.086,\ 0.098)}$ | $\mathbf{0.056}_{(0.052,\ 0.061)}$ |
| RMSE | t-student* | $0.164_{(0.163,\ 0.165)}$ | $0.398_{(0.397,\ 0.400)}$ | $0.215_{(0.198,\ 0.235)}$ | $0.143_{(0.133,\ 0.154)}$ |
| | Laplace* | $0.165_{(0.165,\ 0.166)}$ | $0.399_{(0.397,\ 0.401)}$ | $0.215_{(0.198,\ 0.235)}$ | $0.143_{(0.132,\ 0.153)}$ |
| | Gaussian* | $0.162_{(0.161,\ 0.163)}$ | $0.399_{(0.398,\ 0.401)}$ | $0.209_{(0.193,\ 0.228)}$ | $0.154_{(0.143,\ 0.165)}$ |
| | IQN* | $0.161_{(0.160,\ 0.162)}$ | $0.513_{(0.510,\ 0.515)}$ | $0.210_{(0.194,\ 0.229)}$ | $0.149_{(0.139,\ 0.159)}$ |
| | Informer | $0.324_{(0.322,\ 0.327)}$ | $0.257_{(0.255,\ 0.259)}$ | $0.257_{(0.255,\ 0.259)}$ | $0.307_{(0.305,\ 0.310)}$ |
| | ETSformer | $0.267_{(0.265,\ 0.270)}$ | $0.248_{(0.246,\ 0.249)}$ | $0.249_{(0.247,\ 0.251)}$ | $0.270_{(0.268,\ 0.272)}$ |
| | FEDformer | $0.249_{(0.247,\ 0.251)}$ | $0.216_{(0.214,\ 0.218)}$ | $0.205_{(0.203,\ 0.207)}$ | $0.249_{(0.247,\ 0.251)}$ |
| | Autoformer | $0.235_{(0.233,\ 0.237)}$ | $0.215_{(0.213,\ 0.216)}$ | $0.205_{(0.203,\ 0.206)}$ | $0.239_{(0.237,\ 0.241)}$ |
| | DLinear | $0.084_{(0.083,\ 0.084)}$ | $0.277_{(0.276,\ 0.278)}$ | $\mathbf{0.114}_{(0.107,\ 0.121)}$ | $0.128_{(0.121,\ 0.136)}$ |
| | PatchTST | $0.080_{(0.080,\ 0.081)}$ | $0.241_{(0.239,\ 0.242)}$ | $\mathbf{0.114}_{(0.107,\ 0.121)}$ | $0.094_{(0.087,\ 0.101)}$ |
| | TSMixer | $0.081_{(0.081,\ 0.082)}$ | $0.241_{(0.239,\ 0.242)}$ | $\mathbf{0.108}_{(0.102,\ 0.116)}$ | $0.101_{(0.095,\ 0.109)}$ |
| | tsGT* | $\mathbf{0.075}_{(0.074,\ 0.075)}$ | $\mathbf{0.201}_{(0.200,\ 0.203)}$ | $0.113_{(0.105,\ 0.121)}$ | $\mathbf{0.068}_{(0.063,\ 0.074)}$ |

\* Stochastic model

**Datasets**  To assess the performance of our method against the baseline models we use four real-life datasets, which are commonly used in the field: `electricity`, `traffic`, `ETTm2`, and `weather`, see Wu et al. (2021). The `electricity` and `traffic` datasets are rich in periodic patterns. `ETTm2` introduces additional complexity, as it contains negative values, outliers, and displays unpredictable behavior. Finally, `weather` dataset contains time series with varying scales, frequent outliers, and features a considerable number of zero values in certain series.

Every dataset comprises $S$ time series and each data point is time-stamped. The value $S$ is dataset-dependent, e.g., for `electricity` $S = 321$ and for `traffic` $S = 862$. The training window spans $365 \cdot 24 = 8760$ timesteps, immediately followed by the evaluation horizon of length $H = 24$. See Appendix E for further details.

**Metrics and evaluation protocol**  We evaluate the models on a rolling-window basis. Namely, each dataset is divided into $W = 100$ consecutive and overlapping time windows of equal length. For each time window, we train a separate model and evaluate it on $H$ time steps immediately following that window. This makes the evaluation more robust to outliers and allows us to assess the methods' stability over time. Furthermore, we can compute metrics on a more fine-grained scale and perform backtest. Importantly, this evaluation protocol is well-established in time series literature (Kupiec, 1995; González-Rivera et al., 2004; McNeil et al., 2015), and differs from what is typically used for many transformer baseline models (only one evaluation window and no predictive power assessment, e.g., due to models' determinism). In what follows, we introduce the necessary definitions and describe how we aggregate the results. For additional information, see Appendix F.

For error metrics, we use root-mean-square error (RMSE) and mean-absolute error (MAE), well-established measures that can be computed for deterministic models. For stochastic models, we first compute the mean of the prediction and then compute the error, to achieve fairness in comparison with deterministic models. The metrics are given by the following formulas:

$$\text{RMSE}(w, s) = \sqrt{\frac{1}{H} \frac{1}{F^{s,w}} \sum_{h=1}^{H} \left( \frac{1}{I} \sum_{i=1}^{I} \hat{X}_{h,i}^{s,w} - X_h^{s,w} \right)^2},$$

$$\text{MAE}(w, s) = \frac{1}{H} \frac{1}{F^{s,w}} \sum_{h=1}^{H} \left| \frac{1}{I} \sum_{i=1}^{I} \hat{X}_{h,i}^{s,w} - X_h^{s,w} \right|$$

where $w$, $s$, $H$, and $I$ are the training window, series, horizon length, and number of samples, respectively. Furthermore, $X_h^{s,w}$ is the ground truth target value and $\hat{X}_{h,i}^{s,w}$ is the prediction of a given method, for series $s$, prediction horizon $h$, and simulation $i$. The normalizing factor $F^{s,w}$ is the average absolute value of the ground truth series $s$ in the training window $w$. In this paper, we assume $H = 24$ and $I = 1024$.

For quantile metrics, we report a Quantile Loss (QL) and a Continuous Ranked Probability Score (CRPS) (Zamo & Naveau, 2018; Berrisch & Ziel, 2021). Quantile Loss is defined as

$$\text{QL}_\alpha(w, s) = \frac{2}{H} \frac{1}{F^{s,w}} \sum_{h=1}^{H} \left( \alpha - \mathbf{1}_{\Delta_{h,\alpha}^{s,w} \leq 0} \right) (X_h^{s,w} - \hat{q}_h^{s,w}(\alpha)),$$

where $\hat{q}_h^{s,w}(\alpha)$ is the $\alpha$-quantile predicted by the model for series $s$, prediction horizon $h$, training window $w$, and $\alpha \in (0, 1)$. To compute $\hat{q}_h^{s,w}(\alpha)$ we use an empirical estimator based on $I = 1024$ samples from the model. CRPS summarizes the quantile loss over multiple quantile levels,

$$\text{CRPS}(w, s) = \frac{1}{M} \sum_{m=1}^{M} \text{QL}_{\alpha_m}(w, s),$$

where $\alpha_m = \frac{m}{M+1}$ (we use $M = 20$).

We aggregate error metrics (RMSE, MAE) and quantile metrics (QL, CRPS) for each dataset by summarizing the values over time series and windows via interquartile mean (IQM) (Agarwal et al., 2021). IQM discards bottom and top 25% values and calculates the mean of the remaining 50%, making it robust to outliers. In our experiments, IQM gave results comparable to or more conservative than the median. For results aggregated using an average please see Appendix J.

**Backtest** In addition to the above-mentioned metrics, we use the Kupiec backtest procedure (Kupiec, 1995; McNeil et al., 2015) to measure the model's predictive performance. In particular, how well the model captures the data distribution and the corresponding quantiles. The test is based on the number of quantile violations, defined as:

$$\hat{v}_{h,\alpha}^s = \sum_{w=1}^{W} \mathbf{1}(\hat{q}_h^{s,w}(\alpha) < X_h^{s,w}).$$

If the model correctly models the data randomness, i.e., the test's null hypothesis is true, $\hat{v}_{h,\alpha}^s$ has the Binomial distribution with parameters $(1 - \alpha, W)$, see (McNeil et al., 2015, Section 9.3.1). The likelihood ratio test is given by

$$T_{h,\alpha}^s = 2 \log \frac{Bin(\hat{v}_{h,\alpha}^a, \hat{v}_{h,\alpha}^s/W, W)}{Bin(\hat{v}_{h,\alpha}^s, 1 - \alpha, W)},$$

where $Bin(k, p, n) = \binom{n}{k} p^k (1 - q)^{n-k}$. Under the null hypothesis, $T_{h,\alpha}^s$ is asymptotically distributed as $\chi$-squared distribution with one degree of freedom, $\chi^2(1)$. Consequently, the $p$-value is given as $p\text{-value}_{h,\alpha}^s = \mathbb{P}(\chi^2(1) > T_{h,\alpha}^s)$.

Table 2: Performance of the models in terms of quantile metrics QL and CRPS (lower values are better). Each entry is computed using IQM with 90% bootstrap confidence intervals (in brackets) computed over $W = 100$ training windows and all time series in each dataset.

| metric | model | electricity | traffic | ETTm2 | weather |
|---|---|---|---|---|---|
| CRPS | t-student | 0.11 (0.105, 0.106) | 0.24 (0.235, 0.237) | 0.18 (0.166, 0.198) | 0.12 (0.108, 0.125) |
| | Laplace | 0.11 (0.106, 0.107) | 0.24 (0.234, 0.236) | 0.18 (0.166, 0.199) | 0.12 (0.107, 0.124) |
| | Gaussian | 0.1 (0.103, 0.104) | 0.23 (0.229, 0.23) | 0.17 (0.157, 0.186) | 0.12 (0.114, 0.131) |
| | IQN | 0.1 (0.101, 0.102) | 0.27 (0.271, 0.273) | 0.17 (0.155, 0.185) | 0.11 (0.103, 0.118) |
| | tsGT | **0.04** (0.042, 0.042) | **0.08** (0.081, 0.082) | **0.07** (0.068, 0.078) | **0.04** (0.037, 0.044) |
| $\text{QL}_{50\%}$ | t-student | 0.13 (0.125, 0.126) | 0.27 (0.272, 0.274) | 0.19 (0.172, 0.204) | 0.12 (0.116, 0.134) |
| | Laplace | 0.13 (0.126, 0.127) | 0.27 (0.272, 0.273) | 0.19 (0.171, 0.204) | 0.12 (0.115, 0.134) |
| | Gaussian | 0.12 (0.124, 0.125) | 0.28 (0.274, 0.276) | 0.18 (0.168, 0.199) | 0.14 (0.126, 0.145) |
| | IQN | 0.12 (0.123, 0.125) | 0.34 (0.337, 0.34) | 0.18 (0.168, 0.2) | 0.13 (0.119, 0.136) |
| | tsGT | **0.06** (0.055, 0.056) | **0.1** (0.103, 0.104) | **0.09** (0.086, 0.098) | **0.05** (0.047, 0.056) |
| $\text{QL}_{75\%}$ | t-student | 0.11 (0.11, 0.111) | 0.25 (0.252, 0.254) | 0.17 (0.153, 0.182) | 0.11 (0.102, 0.12) |
| | Laplace | 0.11 (0.111, 0.112) | 0.25 (0.253, 0.255) | 0.17 (0.153, 0.183) | 0.11 (0.101, 0.12) |
| | Gaussian | 0.11 (0.107, 0.108) | 0.25 (0.247, 0.249) | 0.16 (0.146, 0.173) | 0.13 (0.118, 0.137) |
| | IQN | 0.11 (0.108, 0.109) | 0.32 (0.317, 0.32) | 0.16 (0.143, 0.17) | 0.11 (0.102, 0.118) |
| | tsGT | **0.05** (0.046, 0.047) | **0.09** (0.09, 0.091) | **0.08** (0.072, 0.082) | **0.04** (0.04, 0.047) |
| $\text{QL}_{95\%}$ | t-student | 0.06 (0.06, 0.061) | 0.16 (0.162, 0.164) | 0.12 (0.103, 0.129) | 0.06 (0.053, 0.068) |
| | Laplace | 0.06 (0.057, 0.058) | 0.16 (0.155, 0.157) | 0.11 (0.102, 0.128) | 0.06 (0.05, 0.064) |
| | Gaussian | 0.06 (0.056, 0.057) | 0.14 (0.139, 0.14) | 0.09 (0.082, 0.105) | 0.06 (0.051, 0.063) |
| | IQN | 0.05 (0.052, 0.053) | 0.15 (0.152, 0.153) | 0.09 (0.076, 0.098) | 0.05 (0.045, 0.055) |
| | tsGT | **0.02** (0.016, 0.017) | **0.04** (0.037, 0.037) | **0.03** (0.029, 0.035) | **0.02** (0.015, 0.017) |

In experiments, we report the fraction of time series and horizons for which their $p$-value in the underlying likelihood ratio test is at least $\gamma \in (0, 1)$:

$$p\text{-value}_\alpha = \frac{1}{SH} \sum_{h=1}^{H} \sum_{s=1}^{S} \mathbf{1}(p\text{-value}_{h,\alpha}^s \geq \gamma).$$

Informally, this tells us for how many time series and horizons the model correctly predicts tail events according to the Kupiec test with significance level $\gamma$; here $\gamma = 5\%$.

## 5 Experiments

The key finding of this section is that `tsGT` surpasses existing state-of-the-art methods in RMSE and MAE metrics, as detailed in Section 5.1. This highlights the advantage of stochastic models, showing they not only apply to a broader array of scenarios but also provide more accurate pointwise predictions compared to deterministic models. We further analyze the behavior of these models from a distributional perspective in Sections 5.2 to 5.3. It is important to note that such *stochastic analysis is not possible for most of the contemporary baselines due to their determinism*. Our model outperforms stochastic baselines on QL and CRPS (Section 5.2) and does a good job at predicting quantiles (Section 5.3) for datasets `electricity` and `traffic`. However, all models struggle on `ETTm2` and `weather` datasets, highlighting the need for rigorous evaluation and an inflow of new ideas to the field. Finally, we confirm in Section 5.4 that `tsGT` does not suffer from weaknesses pointed out by Zeng et al. (2023). Additional experiments are available in Appendix: for ablations on scaling, normalization, $\beta$, discretization, or embeddings, see Appendix I; for an application of time series forecasting to an atypical task of predicting hand-drawn images, see Appendix H.

Table 3: Fraction of horizon-series pairs in each dataset with $p$-values exceeding 5% for Kupiec's PoF test on each quantile.

| metric | model | electricity | traffic | ETTm2 | weather |
|--------|-------|-------------|---------|-------|---------|
| $p$-value$_{50\%}$ | t-student | 72.7% | **89.9%** | 85.1% | 68.5% |
| | Laplace | 73.7% | 89.5% | 88.1% | 65.1% |
| | Gaussian | 73.2% | 88.6% | 90.5% | **70.4%** |
| | IQN | 74.0% | 46.3% | **94.0%** | 65.5% |
| | tsGT | **88.0%** | 88.9% | 63.1% | 48.8% |
| $p$-value$_{75\%}$ | t-student | 24.6% | 6.7% | 0.0% | 15.3% |
| | Laplace | 22.7% | 6.6% | 0.0% | 14.3% |
| | Gaussian | 36.4% | 21.1% | 5.4% | 17.7% |
| | IQN | 24.1% | 17.0% | 1.8% | 9.7% |
| | tsGT | **78.0%** | **80.7%** | **11.9%** | **40.3%** |
| $p$-value$_{95\%}$ | t-student | 0.7% | 0.0% | 0.0% | 11.9% |
| | Laplace | 1.6% | 0.2% | 0.0% | 13.7% |
| | Gaussian | 1.6% | 0.7% | 0.0% | 17.1% |
| | IQN | 3.0% | 18.8% | 0.0% | 3.4% |
| | tsGT | **67.2%** | **71.8%** | 0.0% | **21.2%** |

## 5.1 Main result: performance on error metrics

Our experiments reveal that `tsGT` outperforms baselines on `electricity`, `traffic`, and `weather` datasets, see Table 1. Importantly, `tsGT`'s confidence intervals do not overlap with those of the other models. On `ETTm2`, `tsGT` matches the performance of DLinear, PatchTST, and TSMixer, as their confidence intervals overlap.

Interestingly, stochastic methods that employ parametric marginal distributions (Gaussian, Laplace, and t-student) often match or surpass the performance of more intricate transformer-based models, e.g., Informer, ETSformer, FEDformer, Autoformer on `electricity`. This observation, especially when combined with further improvement to those results when applying `tsGT`, highlights the value of stochastic modeling in time series forecasting.

We hypothesize that the reasons for the overall `tsGT`'s strong performance are threefold. First, the stochastic nature of `tsGT` allows us to simulate multiple predictions and compute their mean achieving a variance reduction effect. Second, our model benefits from the transformer's dense learning signal in the form of the next token prediction objective. In particular, this incentivizes the emergence of rich internal data representations. Third, `tsGT` models complex non-parametric marginal distributions allowing to capture stochastic behavior of individual time series. This is possible due to our tokenization strategy, which also makes it easy for the transformer to handle real-valued data.

## 5.2 Analysis: quantile metrics

In this section, we investigate the models' performance on quantile metrics QL and CRPS[1], as defined in Section 4. These metrics are useful in ranking the models and the results are presented in Table 2. We observe that `tsGT` outperforms other methods on CRPS by a significant margin. This is encouraging, as this metric summarizes how the model fares in predicting quantiles at multiple levels. The results for QL$_{50\%}$, QL$_{75\%}$, and QL$_{95\%}$ confirm this conclusion. Having ranked the models, we conduct a detailed analysis of their performance in the next section.

---

[1]Such an analysis is unavailable for deterministic baselines considered in this paper, i.e., Informer, ETSformer, FEDformer, Autoformer, DLinear, PatchTST, and TSMixer.

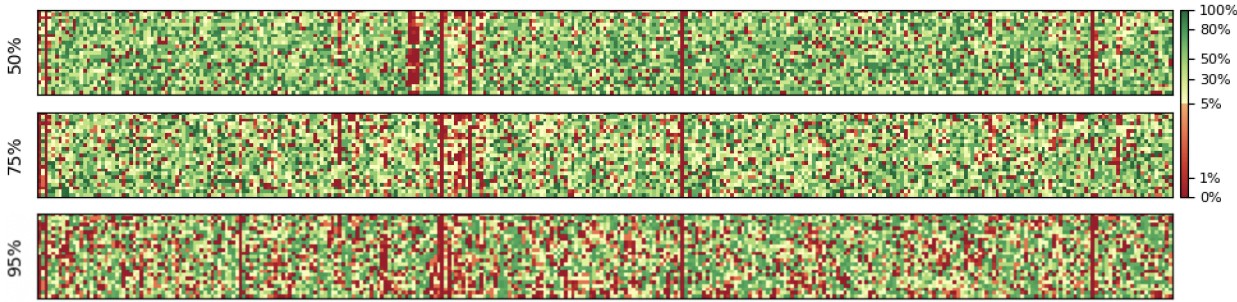

Figure 1: Color-coded p-values from backtesting `tsGT` on `electricity`. Each rectangle represents the results for backtesting `tsGT` on one of the following levels: 50%, 75%, and 95%. The height of a rectangle equals $H = 24$, corresponding to the number of prediction steps, and a width equals to $S = 321$, the number of time series in `electricity`.

### 5.3 Analysis: backtest

In this section, we study the methods' ability to model the data distribution. The findings are presented in Table 3, which reports the outcomes of a backtesting procedure proposed by Kupiec (1995) and described in Section 4 and Appendix F. Namely, the table's entries represent a fraction of all time series and horizons for which the models passed the test. This number summarizes how accurately each model captures the data distribution.

Table 3 indicates that `tsGT` achieves the highest performance at quantile levels 75% and 95% across all datasets. For median, the results vary: `tsGT` leads on `electricity`, matches the performance of t-student, Laplace, and Gaussian on `traffic`, and performs adequately on `weather`, but does not surpass the baselines.

The results presented in Table 3 reveal two key insights. First, there is clearly a negative correlation between $\alpha$ and models' ability to predict $\alpha$-quantile. For many models this drop in performance is severe. Yet, `tsGT` stands out as it consistently maintains good performance across all quantile levels on `electricity` and `traffic`. Second, there is a stark difference in the difficulty of data distribution modeling between datasets. This is particularly visible for `ETTm2` and `weather`, where all the methods achieve good or very good results for modeling the median but collapse for higher quantile levels. We suspect this difficulty stems from the high volatility and unpredictability of time series in these specific datasets.

We notice that the Laplace and t-student approaches do not perform particularly well at 75% and 95% quantile levels on any dataset. This is surprising, given that these distributions are characterized by heavy tails, making them natural candidates for solving the task at hand.

Finally, it is instructive to delve into `tsGT`'s modeling capabilities on a dataset where it excels. Figure 1 showcases p-values from backtesting `tsGT` on `electricity` across three quantile levels ($50\%, 75\%$ and $95\%$), 24 prediction horizons, and 321 individual time series. Each cell in the depicted matrices corresponds to a single p-value, color-coded green for acceptance or red for rejection at the 5% significance level. Green dominates all matrices, indicating a generally high model performance. However, we observe that the amount of red color increases with higher quantile levels, reflecting the rates from Table 3: 88.0% for $\alpha = 50\%$, 78.0% for $\alpha = 75\%$, and 67.2% for $\alpha = 95\%$. Additionally, we can see that for certain time series, predicting quantiles across any horizon remains particularly challenging, as indicated by the vertical red lines.

### 5.4 Analysis: input permutation

Transformer models, including Informer, Autoformer, and FEDformer have been criticized for insufficient handling of temporal dependencies, and some even exhibit permutational invariance (Zeng et al., 2023). This can be viewed as a serious issue, as the sequential order often plays a crucial role in time series modeling. Additionally, Zeng et al. (2023) demonstrated that these methods underperform with respect to a simple

Table 4: Performance of `tsGT` after a random shuffling of input during inference.

| metric | model | electricity | traffic | ETTm2 | weather |
|---|---|---|---|---|---|
| MAE | tsGT + shuffling | $0.283_{(0.282, 0.285)}$ | $0.590_{(0.589, 0.592)}$ | $1.306_{(1.199, 1.426)}$ | $0.210_{(0.197, 0.225)}$ |
| | tsGT | $0.057_{(0.057, 0.058)}$ | $0.114_{(0.114, 0.115)}$ | $0.092_{(0.086, 0.098)}$ | $0.056_{(0.052, 0.061)}$ |
| RMSE | tsGT + shuffling | $0.328_{(0.327, 0.329)}$ | $0.733_{(0.731, 0.735)}$ | $1.398_{(1.291, 1.517)}$ | $0.239_{(0.224, 0.255)}$ |
| | tsGT | $0.075_{(0.074, 0.075)}$ | $0.201_{(0.200, 0.203)}$ | $0.113_{(0.105, 0.121)}$ | $0.068_{(0.063, 0.074)}$ |

MLP-based model, DLinear. We show that `tsGT` suffers a significant drop in performance when randomly shuffling the input (Table 4) and that its performance is superior to DLinear (Table 1), addressing the main concerns of (Zeng et al., 2023) about using transformers for time series modeling.

## 6 Limitations and future work

**Quadratic time and memory complexity of self-attention**  This well-known issue of transformers negatively impacts the options for making long-horizon predictions. However, due to the generality of our approach, any progress in this area, which is an active field of research (Tay et al., 2022; Zhou et al., 2024; Tang et al., 2024), can be directly applied to `tsGT` and is left as a future work.

**Backtesting**  Backtesting is an effective tool to measure the predictive performance of the model, particularly in situations when adequate risk capturing is of special importance. For large transformer models, this can be computationally expensive. However, a low-rank adaptation technique (Hu et al., 2021) holds some promise in alleviating this problem.

**Access to marginal probability distribution function**  Generic transformer models allow sampling trajectories but do not give easy access to the probability distribution functions. It would be interesting to see how this obstacle can be overcome.

**Uncertainty-aware model**  It would be beneficial to study whether `tsGT` can evaluate its own uncertainty, similarly to (Kadavath et al., 2022).

**Normalization**  Normalization has some non-trivial interactions with digit distribution (Hill, 1995) and in the future work, we would like to test other recently developed normalization schemes (Kim et al., 2022; Nie et al., 2022).

**Explainability**  Reverse engineering transformer models remains a challenging task. Nevertheless, recent advancements in the so-called mechanistic interpretability, see (Nanda et al., 2023; Chughtai et al., 2023; Wang et al., 2023), offer promising approaches to address this issue. Investigating the underlying circuits of learned time series transformers is an exciting avenue for future research.

## 7 Conclusions

In this paper, we propose `tsGT`, a general-purpose transformer stochastic time series model. We focus on evaluating the model's predictive capabilities, employing a rolling window analysis, a well-known time-series procedure. We show that `tsGT` outperforms the current state-of-the-art models on MAE and RMSE on four popular datasets: `electricity`, `traffic`, `ETTm2`, `weather`. Similarly, we demonstrate that `tsGT` surpasses its stochastic peers on QL and CRPS. We complement these results with a thorough analysis using the Kupiec backtest. We show how the models behave on datasets of varying complexity and at different quantile levels. We go as deep as showing the test results for each prediction period, quantile level, and time series in `electricity`. We close the analysis showing that `tsGT` is not permutationally invariant. This addresses a prominent critique directed towards some contemporary time series transformer models, raised by (Zeng et al., 2023). Appendix contains results of additional ablation studies.

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

# A   Model's architecture

## A.1   Transformer

The transformer (Vaswani et al., 2017) is a deep neural network model that relies on the attention mechanism. We employ a decoder-only variant of this architecture, a common choice in natural language modeling (Radford et al., 2019; Brown et al., 2020). A transformer model takes a sequence of tokens as input and outputs a sequence of logits over all the tokens. The transformer's central object is a residual stream, which serves as a working memory for the model. Each token undergoes the initial embedding into a space of dimension $d_{\text{model}}$, after which it is processed by $N$ decoder layers. These layers consist of attention and feed-forward (FF) blocks. These blocks can be conceptualized as operations on the residual stream. Specifically, each one of $n_{\text{heads}}$ heads inside the attention block operates on dimension $d_{\text{head}} = d_{\text{model}}/n_{\text{heads}}$ and moves information between positions, while the FF block writes or removes from tokens' residual streams (Nanda et al., 2023). The FF block incorporates nonlinearity (we use the Gaussian Error Linear Unit or GELU) and is composed

of a two-layer multilayer perceptron (MLP) with a hidden space of dimension $d_{\text{ff}}$. The inputs to attention, feed-forward blocks, and the output logits layer are processed by layer norms.

The causal self-attention mechanism for each element of the sequence assigns different weights to each preceding token based on its relevance to the current element, which allows for flexible and context-aware representations. Independent attention operations are executed $H$ times, once for every attention head, enabling the model to learn distinct representations. For each head, the attention function can be expressed as:

$$\text{Attention}(X) = \text{softmax}\left(\frac{QK^T + M}{\sqrt{d_{\text{head}}}}\right) V,$$

where $Q = XW^Q$, $K = XW^K$ and $V = XW^V$ for $W^Q$, $W^K$, $W^V \in \mathbb{R}^{d_{\text{model}} \times d_{\text{head}}}$ are trainable linear projections of the residual stream normalized by layer norm, and $M$ is a causal mask with zeros below the diagonal and negative infinity elsewhere. The outputs of attention heads are concatenated and linearly projected to $d_{\text{model}}$. We also extend the attention mechanism by incorporating rotary encodings (Su et al., 2021). For more details on the transformer architecture, please refer to the original work of Vaswani et al. (2017).

### A.2 Data normalization

Here we describe the data normalization procedure, i.e., step (a) described in the input and output discretization part of Section 3. Inspired by (Salinas et al., 2020), we scale model input $X$ by dividing it by $\mu_X = r + \frac{1}{T}\sum_{t=1}^{T}|X_t|$, where $T$ denotes the sequence length and $r$ is a positive constant. We invert the scaling at the model's output. During prediction, we calculate the scaling factor based only on the context to prevent information leaks from the prediction horizon.

### A.3 Data quantization and discretization into digits

---
**Algorithm 1** Digit Factorization
---

| **Requires:** | $x$ | normalized input (see Appendix A.2) to be discretized into $p$ digits in base $v$ |
| | $v$ | vocabulary size, which acts as the number base |
| | $p$ | precision to be used in the discretization |
| | $l, h$ | lower and upper bounds used for squashing $x$ into $[0, 1]$ |

1: **function** DIGITFACTORIZATION($x$; $p$, $v$, $l$, $h$)
2:     $x \leftarrow \text{clip}((x - l)/(h - l), 0, 1)$                    ▷ Squash to $[0, 1]$
3:     $D \leftarrow []$
4:     $r \leftarrow 1$
5:     **for** $i = 1$ to $p$ **do**
6:         $r \leftarrow r/v$
7:         $d \leftarrow \min(\lfloor x/r \rfloor, v - 1)$
8:         $D.\text{append}(d)$
9:         $x \leftarrow x - d \cdot r$
10:    **return** $D$

---

The input to the transformer decoder (Appendix A.1) is normalized (Appendix A.2) and discretized into digits, see Algorithm 1. The selected number base $v$ equals the vocabulary size used for the model. We squash the input $x$ into $[0, 1]$ interval and extract $p$ most significant digits. Hence, a sequence of length $T$ is discretized into a sequence of $p \cdot T$ digits. We notice that the higher the precision, the longer the input. Put differently, with a fixed context length, increasing $p$ reduces the amount of raw data fed into the model. This is a limitation caused by the quadratic complexity of transformers, see also Section 6.

For datasets that do not contain negative values, i.e., `electricity` and `traffic`, we set the parameters $l$ and $h$ in Algorithm 1 to 0 and 10, respectively. For datasets `ETTm2` and `weather`, which contain negative values, we set $l = -10$ and $h = 10$.

## B  Hyperparameters

### B.1  Hyperparameters of baselines

The hyperparameters used to configure and train the baselines are given in Table 5 and Table 6. The parameters were chosen according to configurations supplied in the respective public repositories, except for `pred_len`, `seq_len` and `label_len`, which were adjusted to match the training and evaluation setup of this work.

Table 5:  Hyperparameters of baselines. Names refer to parameters in the corresponding code repositories.

|  | Autoformer | ETSformer | FEDformer | Informer | PatchTST | DLinear | TSMixer |
|---|---|---|---|---|---|---|---|
| d_model | 512 | 512 | 512 | 512 | 128 | — | 64 |
| d_ff | 2048 | 2048 | 2048 | 2048 | 256 | — | 32-64 |
| activation | gelu | gelu | gelu | gelu | gelu | gelu | relu |
| e_layers / n_block | 2 | 2 | 2 | 2 | 3 | — | 2 - 8 |
| d_layers | 1 | 1 | 1 | 1 | — | — | — |
| n_heads | 8 | 8 | 8 | 8 | 16 | — | — |
| seq_len | 232 | 232 | 232 | 232 | 232 | 232 | 232 |
| pred_len | 24 | 24 | 24 | 24 | 24 | 24 | 24 |
| label_len | 116 | 0 | 116 | 116 | 48 | 48 | — |
| distilling in encoder | True | True | True | True | True | — | — |
| train_epochs | 3-10 | 3-10 | 3-10 | 3-10 | 100 | 100 | 100 |
| batch_size | 32 | 32 | 32 | 32 | 24 - 128 | 16 - 32 | 32 |
| learning_rate | 1e-4 | 1e-5 - 1e-3 | 1e-4 | 1e-4 | 1e-4 | 1e-4 - 5e-2 | 1e-4 - 1e-3 |
| dropout | 0.05 | 0.05 | 0.05 | 0.05 | 0.2 | — | 0.3 - 0.9 |
| loss | mse | mse | mse | mse | mse | mse | mse |
| lradj | type1 | type1 | type1 | type1 | TST, type3 | type3 | — |
| ES patience | 3 | 3 | 3 | 3 | 10 - 20 | — | 5 |
| RevIn | — | — | — | — | True | — | True |

Table 6: Hyperparameters of baseline models which are adjusted per-dataset.

| model | parameter | electricity | traffic | ETTm2 | weather |
|---|---|---|---|---|---|
| Autoformer | train_epochs | 10 | 3 | 10 | 10 |
| ETSformer | train_epochs | 10 | 3 | 10 | 10 |
|  | K | 3 | 3 | 3 | 1 |
|  | learning_rate | 3e-4 | 1e-3 | 1e-5 | 1e-3 |
| FEDformer | train_epochs | 10 | 3 | 10 | 10 |
| Informer | train_epochs | 10 | 3 | 10 | 10 |
| PatchTST | early stopping patience | 10 | 10 | 20 | 20 |
|  | batch_size | 32 | 24 | 128 | 128 |
| DLinear | learning_rate | 1e-3 | 5e-2 | 1e-4 | 1e-3 |
|  | batch_size | 16 | 16 | 16 | 32 |
| TSMixer | learning_rate | 1e-4 | 1e-4 | 1e-3 | 1e-4 |
|  | n_block | 4 | 8 | 2 | 4 |
|  | d_ff | 64 | 64 | 64 | 32 |
|  | dropout | 0.7 | 0.7 | 0.9 | 0.3 |

### B.2 Hyperparameters of `tsGT`

We performed a grid search over the number of layers $(4, 6, 8)$, multifactor constant which controls the learning rate $(0.01, 0.03, 0.1)$, the number of attention heads $(2, 4, 8)$, dimension of model $(64, 128, 256, 512)$, dropout $(0.1, 0.2, 0.3, 0.4, 0.5)$, precision $(1, 2, 3)$, and vocabulary size $(1024, 512, 10)$. Based on this grid search, we chose a universal set of hyperparameters which is summerized in Table 7.

Table 7: Hyperparameters for `tsGT`

| | |
|---:|:---|
| $d_{\mathrm{model}}$ | 256 |
| $d_{\mathrm{ff}}$ | 512 |
| activation | gelu |
| $N$ | 6 |
| $n_{\mathrm{heads}}$ | 4 |
| $T$ | 256 |
| $H$ | 24 |
| train_steps | $10^5$ |
| batch_size | 16 |
| multifactor constant | 0.03 |
| dropout | 0.1 |
| precision | 3 |
| vocab_size | 10 |

## C  Stochastic baselines

### C.1  Distribution-based baselines

As a stochastic baseline for our model, inspired by Li et al. (2019), we implement models that predict the parameters of the probability distributions. We use a transformer backbone and train the models to maximize the log probability of training data. We compare against 3 different symmetric distributions: t-student, Laplace, and Gaussian. For all those, we train the scale and location parameters of the distribution. We use training pipeline based on next token prediction. The hyperparameters of the model and the optimizer are fixed. They are kept the same as those used for `tsGT` (see Table 7).

Similar to Li et al. (2019) we use convolutional attention, and set the kernel width to 3, after running a hyperparameter search. We also use additional covariates. All models use series index, a logarithm of the timestep, and an hour, as well as either a day in case of `electricity` and `traffic` datasets, or a minute in case of `ETTm2` and `weather`.

### C.2  Implicit Quantile Baseline

Another stochastic baseline is inspired by Gouttes et al. (2021). We train a transformer architecture with Implicit Quantile Module (IQM) that learns the inverse of the cumulative probability function using Quantile Loss. Again, the hyperparameters of the Transformer backbone and training hyperparameters are fixed. They are kept the same as those used to tune `tsGT` (see Table 7). The only hyperparameter left is the width of the embedding in the IQM head, which is kept at 256 to match the size of the hidden representation of the transformer. We train a transformer with IQM head to perform next token prediction, by minimizing quantile loss. For more details on the IQM and its exemplar application for time series modeling please refer to Gouttes et al. (2021).

## D  Deterministic baselines

Among others, we used the following methods as baselines: FEDformer (Zhou et al., 2022), Informer (Zhou et al., 2021), Autoformer (Wu et al., 2021), ETSformer (Woo et al., 2022).

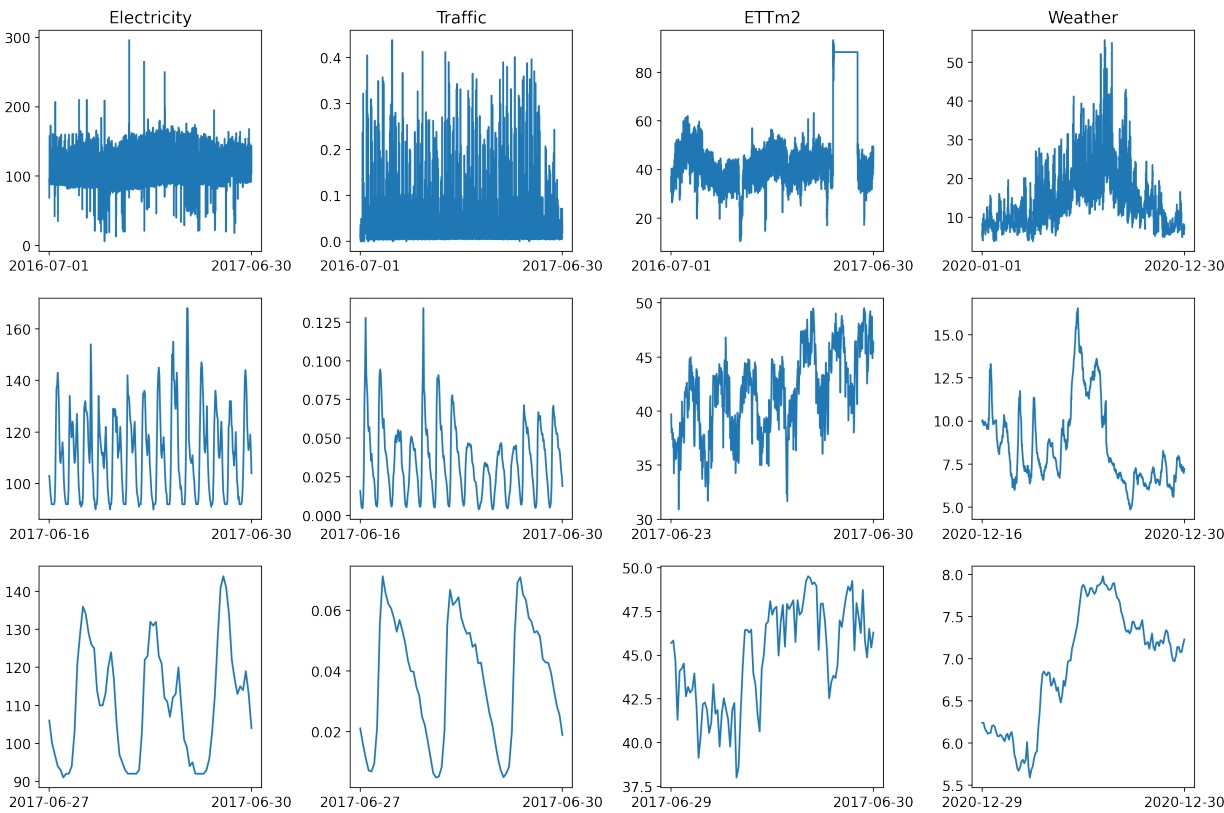

Figure 2: Visualization of data from each of the four real-life datasets. The first row displays a year's worth of data for a selected series. The second and the third rows zoom in on the suffix of those trajectories.

All the above-mentioned baseline models, except for ETSformer, include additional data to the input; these are composed of timestamps (e.g., minute, hour, week, month, and year) or other types of information (e.g., holidays and events). This is done by adding additional learnable encoding to the architecture. The baselines deterministically output a prediction trajectory.

Each model proposes some efficiency improvements to reduce the quadratic memory complexity of the transformer: (Li et al., 2019) proposes log-sparse attention while (Zhou et al., 2021; 2022; Wu et al., 2021) use low-rank property of attention.

Furthermore, each baseline implements a time series-specific architectural change: (Li et al., 2019) introduces 1d-conv attention, (Zhou et al., 2021) designs a decoder to produce prediction trajectories directly, (Wu et al., 2021) introduces a seasonal-trend decomposition with an auto-correlation block, (Zhou et al., 2022; Woo et al., 2022) explore attention mechanisms in the frequency domain.

## E   Data

We use four real-life datasets popular in the field: `electricity`, `traffic`, ETTm2, and `weather`, see Wu et al. (2021) and the references therein. Exemplary trajectories are visualized in Figure 2, and datasets statistics are provided in Table 8.

The `electricity` dataset records the hourly electricity consumption of 321 customers from July 2016 to July 2019. The `traffic` dataset comprises hourly Caltrans data from July 2016 to July 2018 on road occupancy rates for San Francisco Bay area freeways, measured by 862 sensors. Both `electricity` and `traffic` are characterized by a large number of periodic patterns.

| Dataset | Total length | Number of sequences | Frequency |
|---|---|---|---|
| electricity | 26,304 | 321 | 1H |
| traffic | 17,544 | 862 | 1H |
| ETTm2 | 69,680 | 7 | 15T |
| weather | 52,704 | 21 | 10T |

Table 8: Statistics of the datasets used in this paper.

The `ETTm2` dataset was collected from electricity transformers in a province of China and contains seven time series representing, e.g., load and oil temperature recorded every 15 minutes between July 2016 and July 2018. It includes negative values, outliers, and erratic behavior.

The `weather` dataset holds the recordings of 21 meteorological indicators collected every 10 minutes between January 2020 and January 2021 in Germany. The data includes air temperature or humidity and is characterized by varying scales, outliers, and a significant number of zero values.

## F  Metrics

**Error metrics**  Error metrics are defined in the paragraph 'Metrics and evaluation protocol' in Section 4. The normalizing constant $F^{s,w}$, for $w \in \{1, \ldots, W\}$ and $s \in \{1, \ldots, S\}$ is defined as $F^{s,w} = \frac{1}{T^w_{end} - T^w_{start}} \sum_{k=T^w_{start}}^{T^w_{end}} |X^{s,w}_k|$, where $T^w_{start}, T^w_{end}$ are the training window's $w$ beginning and the end, and $X^{s,w}$ is the ground truth series.

**Backtest metrics**  We use the Kupiec proportion of failures (POF) likelihood ratio test (Kupiec, 1995) to backtest the model. We use $W = 100$ train windows and train the model on each, starting from randomly initialized weights. Denote by $X^{s,w}_h$ the ground truth target value for series $s \in \{1, \ldots, S\}$, prediction horizon $h \in \{1, \ldots, H\}$, and training window $w \in \{1, \ldots, W\}$. Let $\alpha$ be the confidence level and denote the corresponding quantile estimate of the model for the $h$-step ahead prediction in window $w$ and series $s$, by $\hat{q}^{s,w}_h(\alpha)$. Then, the number of model failures to correctly predict the quantile can be measured by a statistics $\hat{v}^s_{h,\alpha}$ defined as:

$$\hat{v}^s_{h,\alpha} = \sum_{w=1}^{W} \mathbf{1}(\hat{q}^{s,w}_h(\alpha) < X^{s,w}_h).$$

If the model is correct (that is, under the null hypothesis), $\hat{v}^s_{h,\alpha}$ has the Binomial distribution with parameters $(1 - \alpha, W)$, see (McNeil et al., 2015, Section 9.3.1). The likelihood ratio test is given by

$$T^s_{h,\alpha} = 2 \log Bin(\hat{v}^a_{h,\alpha}, \hat{f}^s_{h,\alpha}, W) - 2 \log Bin(\hat{v}^s_{h,\alpha}, 1 - \alpha, W),$$

where $\hat{f}_{h,\alpha} \triangleq \hat{v}^s_{h,\alpha}/W$ and $Bin(k, p, n) = \binom{n}{k} p^k (1 - q)^{n-k}$. Under the null hypothesis, $T^s_{h,\alpha}$ is asymptotically distributed as $\chi$-squared distribution with one degree of freedom, $\chi^2(1)$.

Correspondingly, the $p$-value is computed as

$$p\text{-value}^s_{h,\alpha} = \overline{F}_{\chi^2(1)}(T^s_{h,\alpha}),$$

where $F_{\chi^2(1)} = 1 - \overline{F}_{\chi^2(1)}$ is the cumulative distribution function of $\chi^2(1)$. We report a proportion of all time series for which the p-value is at least $\gamma$, i.e.,

$$p\text{-value}_\alpha = \frac{1}{SH} \sum_{h=1}^{H} \sum_{s=1}^{S} \mathbf{1}(p\text{-value}^s_{h,\alpha} \geq \gamma),$$

where $S$ denotes the number of time series in the dataset.

## G  Computational infrastructure, cost, and source code

We conducted our experiments using a cluster with Nvidia A100 graphics cards. Each training used a single GPU card, up to 128GB of RAM, and 16 CPU cores. Training our model took between 30 minutes and 2 hours, depending on the number of time series in the dataset. The simulation phase (when we sampled 1024 predictions) took up to 3.5 hours. We estimate the overall cost of reproducing the results to be 2000 GPUh.

The project's overall cost, including development, prototyping, and evaluation of our method and benchmarking the baseline methods, took approximately 80K GPUh.

## H  The Quick, Draw! Dataset

To check how our method performs on tasks, which do not involve 'traditional' time series, we trained `tsGT` on apple sketches of The Quick, Draw! Dataset[2]. This dataset is a collection of 50 million drawings divided into 345 categories, gathered from players of the game Quick, Draw!. In this experiment, we treat sketches as two-dimensional $(x, y)$ time series, an extension to our univariate input and output discretization schema from Section 3.

When modelling a $d$-variate time series, we consider $\mathbb{X}_{t-T+1:t} = (X_{t-T+1}, \ldots, X_t)$, where now $X_t \in \mathbb{R}^d$. A time series $\mathbb{X}$ is translated into a sequence of tokens, by first flattening it into a one-dimensional sequence of intertwined series, which has a length $d \cdot T$. Each value in such sequence is then broken down into digits, as described in Section 3. This results in a series of tokens $\tau(\mathbb{X}) = (\tau_t(\mathbb{X}))_{t=1}^{p \cdot d \cdot T}$ of length $p \cdot d \cdot T$. After processing it with `tsGT`, tokens are generated one by one and gathered into groups of size $p \cdot d$ which are later reinterpreted as distinct time steps of a multivariate series.

In evaluation, we remove and reconstruct a part of the image with the model. Selected examples are shown in Figure 3. We observe that, in general, the reconstructions are plausible and faithfully depict high-level features (e.g., the shape is 'closed'). On the negative side, the model struggles with low-level details. Samples of `tsGT` reconstructing images from the banana, castle, or flower classes are given in Figure 4, Figure 5, and Figure 6, respectively.

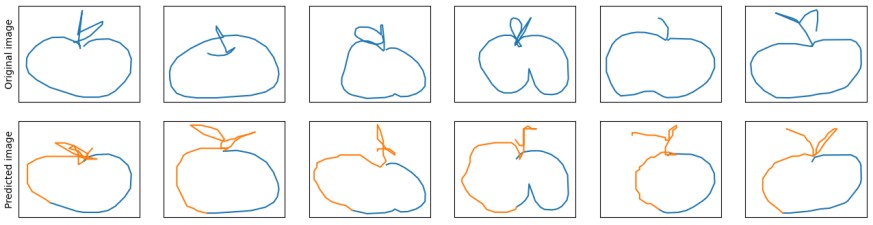

Figure 3: Reconstruction of apple sketches by `tsGT`. The ground truth images are shown in the top row. In the bottom row, the partial ground truth trajectory used to prompt `tsGT` is shown in blue, and the reconstructed trajectory is highlighted in orange.

## I  Ablations

In this section, we present several ablation studies. We analyze different design choices for scaling the raw time series input, normalization to $[0, 1]$, embeddings, and significance weighting. We conducted the experiments on `electricity` dataset.

### I.1  Scaling the raw time series input

Before data is fed to the transformer, it is scaled. `tsGT` performs that by dividing the input sequence of length $T$ by its mean (following Li et al. (2019)). This approach, however, leaks data in the sense that each

---
[2]`https://github.com/googlecreativelab/quickdraw-dataset`

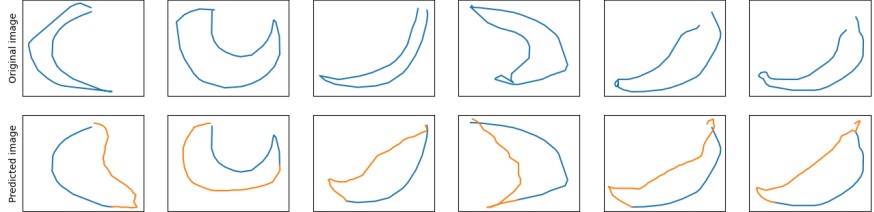

Figure 4: Reconstruction of banana sketches by `tsGT`. The ground truth images are shown in the top row. In the bottom row, the partial ground truth trajectory used to prompt `tsGT` is shown in blue, and the reconstructed trajectory is highlighted in orange.

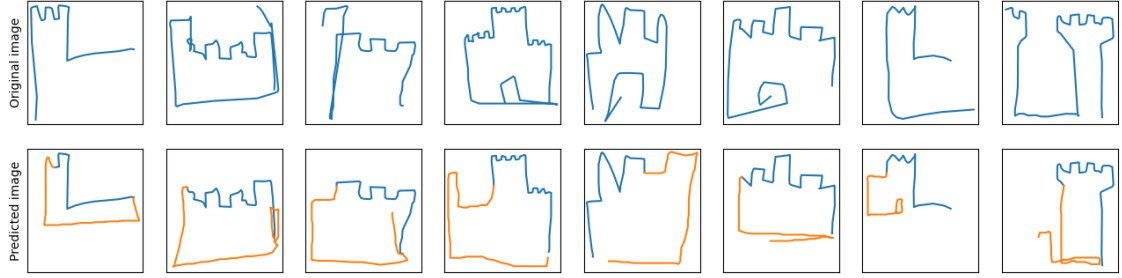

Figure 5: Reconstruction of castle sketches by `tsGT`. The ground truth images are shown in the top row. In the bottom row, the partial ground truth trajectory used to prompt `tsGT` is shown in blue, and the reconstructed trajectory is highlighted in orange.

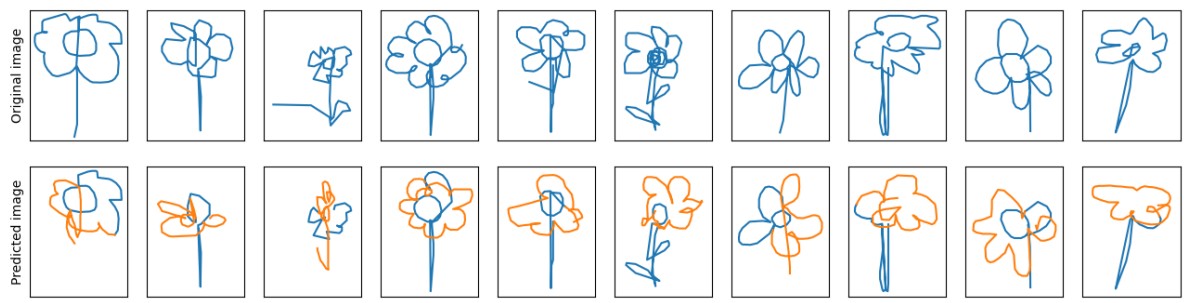

Figure 6: Reconstruction of flower sketches by `tsGT`. The ground truth images are shown in the top row. In the bottom row, the partial ground truth trajectory used to prompt `tsGT` is shown in blue, and the reconstructed trajectory is highlighted in orange.

token is scaled by the statistic dependent on the whole context. Theoretically, the model could exploit this information during training at the expense of its performance at prediction time when it suffers a distribution shift by only having access to the mean of $T - H$ values. To check whether this matters in practice, we implemented a causal normalization procedure, where each value in the sequence is normalized only by the mean of the preceding input prefix. Using a well-known recursive property of rolling mean avoids information leaks and, consequently, distribution shift at inference time. We present the results of the comparison in Table 9. Somewhat surprisingly, the only significant difference between the approaches is on QL-99, where the non-causal approach achieves better results. This might suggest that the data leak problem is not too severe or that the other data preprocessing steps mitigate the problem. The non-causal approach used in `tsGT` is also conceptually simpler and easier to code.

Table 9: Comparison of different normalization methods on `electricity` dataset. The presented values are Inter Quantile Means computed over 10 training windows and 321 training time series. 95% bootstrap confidence interval is provided in brackets.

| model | RMSE | MAE | QL-99 | QL-95 |
|---|---|---|---|---|
| `tsGT` per-ts norm. | $0.062_{(0.058,\ 0.067)}$ | $0.047_{(0.044,\ 0.050)}$ | $0.004_{(0.004,\ 0.004)}$ | $0.014_{(0.013,\ 0.015)}$ |
| `tsGT` causal norm. | $0.063_{(0.058,\ 0.068)}$ | $0.048_{(0.044,\ 0.051)}$ | $0.005_{(0.004,\ 0.005)}$ | $0.015_{(0.014,\ 0.016)}$ |

### I.2 Normalization to $[0, 1]$

Appendix A.3 describes the scaling of the normalized input data to $[0, 1]$. `tsGT` follows Li et al. (2019) in reducing the range of data by dividing by a constant factor (10 in our case) and clipping the outcome to $[0, 1]$. Note that this procedure clips all the negative values to 0, which is undesirable for datasets containing negative values. Instead, for such datasets (`ETTm2`, `weather`) we add a constant value of 10 to the time series values, divide by 20, and clip to range $[0, 1]$.

Alternatively, we tested a sigmoid function for softer scaling. Intuitively, this approach could preserve outliers better as it provides additional space for samples far from the average value. The results of the comparison can be found in Table 10. The metric values are higher for the approach used by `tsGT`, though the approaches are comparable in statistical terms.

Table 10: Comparison of different preprocessing methods on `electricity` dataset. The presented values are Inter Quantile Means computed over 10 training windows and 321 training time series. 95% bootstrap confidence interval is provided in brackets.

| model | RMSE | MAE | QL-99 | QL-95 |
|---|---|---|---|---|
| `tsGT` clip | $0.062_{(0.058,\ 0.067)}$ | $0.047_{(0.044,\ 0.050)}$ | $0.004_{(0.004,\ 0.004)}$ | $0.014_{(0.013,\ 0.015)}$ |
| `tsGT` squash | $0.064_{(0.059,\ 0.069)}$ | $0.048_{(0.045,\ 0.052)}$ | $0.004_{(0.004,\ 0.004)}$ | $0.014_{(0.013,\ 0.015)}$ |

### I.3 Significance weighting $\beta$

To train `tsGT`, we use a weighted loss, see equation 1. The parameter $\beta$ controls the contribution of each digit to the loss according to its significance. We compare the performance of different values of $\beta$, see Table 11. In most cases, the metrics for $\beta = 0.3$ (used in `tsGT`) are the most favorable, though from a statistical perspective, the outputs are rather stable under different choices of $\beta$.

### I.4 Discretization and embeddings

Normalized real values are discretized before being passed to `tsGT`, as described in Section A.2. When we fix the normalization factor, the function that assigns the most significant digit to raw values splits the space of raw inputs $\mathbb{R}$ into $B$ regions. By the linearity of transformations, those regions, apart from the two outermost ones, form intervals of the same length. We could consider other discretization schemes, which assign the

Table 11: Comparison of different $\beta$ values on `electricity` dataset. The presented values are Inter Quantile Means computed over 10 training windows and 321 training time series. 95% bootstrap confidence interval is provided in brackets.

| model | RMSE | MAE | QL-99 | QL-95 |
|-------|------|-----|-------|-------|
| `tsGT` $\beta$=0.3 | $0.062_{(0.058,\ 0.067)}$ | $0.047_{(0.044,\ 0.050)}$ | $0.004_{(0.004,\ 0.004)}$ | $0.014_{(0.013,\ 0.015)}$ |
| `tsGT` $\beta$=0.6 | $0.065_{(0.061,\ 0.071)}$ | $0.050_{(0.046,\ 0.054)}$ | $0.004_{(0.004,\ 0.005)}$ | $0.015_{(0.014,\ 0.016)}$ |
| `tsGT` $\beta$=0.9 | $0.064_{(0.059,\ 0.069)}$ | $0.048_{(0.045,\ 0.052)}$ | $0.004_{(0.004,\ 0.004)}$ | $0.014_{(0.013,\ 0.015)}$ |
| `tsGT` $\beta$=0.99 | $0.065_{(0.061,\ 0.069)}$ | $0.049_{(0.047,\ 0.052)}$ | $0.004_{(0.003,\ 0.004)}$ | $0.014_{(0.013,\ 0.015)}$ |

Table 12: Comparison of different approaches to discretization and digit embedding. "Quantile discretization" refers to a discretization schema in which the most significant digits appear with the similar frequency. "Separate embeddings" refers to the experiment in which the same digits appearing at different positions in a representation of a number are assigned different vectors in the embedding space.

| metric | model | electricity | ETTm2 |
|--------|-------|-------------|-------|
| MAE | `tsGT` – quantile discretization | $0.060_{(0.059,\ 0.060)}$ | $0.095_{(0.088,\ 0.102)}$ |
| | `tsGT` – separate embeddings | $0.062_{(0.061,\ 0.062)}$ | $0.092_{(0.087,\ 0.099)}$ |
| | `tsGT` | $0.057_{(0.057,\ 0.058)}$ | $0.092_{(0.086,\ 0.098)}$ |
| RMSE | `tsGT` – quantile discretization | $0.078_{(0.077,\ 0.078)}$ | $0.116_{(0.107,\ 0.125)}$ |
| | `tsGT` – separate embeddings | $0.080_{(0.079,\ 0.080)}$ | $0.114_{(0.107,\ 0.123)}$ |
| | `tsGT` | $0.075_{(0.074,\ 0.075)}$ | $0.113_{(0.105,\ 0.121)}$ |

most significant digits differently. In particular, we consider a function which splits the raw input space into $B$ uneven intervals, such that those intervals contain a similar number of observed values. Intuitively, instead of evenly splitting the input space, we split evenly along the quantiles of empirical distribution of time series values. We present the results of this ablation in Table 12. On the two datasets on which we run this ablation, we find no advantage to using this method.

Our model uses the same tokens to represent digits of different significance. Potentially, using the exact same embeddings coupled with the distributional differences between tokens appearing on different positions could impact models performance. We check, if allowing `tsGT` to learn different embeddings for digits appearing on different positions could improve its performance, but find that this is not the case, as evident from Table 12. We hypothesize that out method is able to capture temporal dependencies well enough to distinguish the position in which a digit appears and builds common internal representation of numbers.

## J    Additonal experimental results

Table 13: Performance of models in terms of error metrics MAD and RMSE (lower values are better). Each entry is an average and accompanied by a 90% bootstrap confidence interval (in brackets) computed over $W = 100$ training windows and all time series.

| metric | model | electricity | traffic | ETTm2 | weather |
|---|---|---|---|---|---|
| MAD | t-student* | $0.165_{(0.163,\ 0.169)}$ | $0.314_{(0.313,\ 0.315)}$ | $0.465_{(0.403,\ 0.534)}$ | $0.392_{(0.316,\ 0.492)}$ |
| | Laplace | $0.167_{(0.164,\ 0.170)}$ | $0.314_{(0.313,\ 0.316)}$ | $0.489_{(0.409,\ 0.587)}$ | $0.394_{(0.320,\ 0.493)}$ |
| | Gaussian | $0.165_{(0.162,\ 0.168)}$ | $0.315_{(0.314,\ 0.316)}$ | $0.467_{(0.395,\ 0.552)}$ | $0.405_{(0.330,\ 0.507)}$ |
| | IQN | $0.166_{(0.163,\ 0.170)}$ | $0.418_{(0.416,\ 0.420)}$ | $0.455_{(0.389,\ 0.530)}$ | $0.428_{(0.342,\ 0.537)}$ |
| | Informer | $0.277_{(0.275,\ 0.280)}$ | $0.242_{(0.240,\ 0.244)}$ | $0.245_{(0.243,\ 0.247)}$ | $0.272_{(0.266,\ 0.279)}$ |
| | ETSformer | $0.260_{(0.257,\ 0.264)}$ | $0.242_{(0.240,\ 0.244)}$ | $0.243_{(0.241,\ 0.246)}$ | $0.262_{(0.256,\ 0.269)}$ |
| | FEDformer | $0.222_{(0.220,\ 0.225)}$ | $0.206_{(0.205,\ 0.208)}$ | $0.201_{(0.198,\ 0.203)}$ | $\mathbf{0.228}_{(\mathbf{0.222},\ \mathbf{0.235})}$ |
| | Autoformer | $0.213_{(0.211,\ 0.216)}$ | $0.208_{(0.206,\ 0.210)}$ | $0.203_{(0.200,\ 0.205)}$ | $\mathbf{0.223}_{(\mathbf{0.218},\ \mathbf{0.230})}$ |
| | PatchTST | $0.089_{(0.088,\ 0.091)}$ | $0.193_{(0.192,\ 0.194)}$ | $\mathbf{0.175}_{(\mathbf{0.157},\ \mathbf{0.196})}$ | $0.313_{(0.239,\ 0.414)}$ |
| | DLinear | $0.095_{(0.094,\ 0.097)}$ | $0.220_{(0.219,\ 0.221)}$ | $\mathbf{0.174}_{(\mathbf{0.156},\ \mathbf{0.194})}$ | $0.351_{(0.277,\ 0.450)}$ |
| | TSMixer | $0.096_{(0.091,\ 0.102)}$ | $0.192_{(0.191,\ 0.193)}$ | $\mathbf{0.168}_{(\mathbf{0.150},\ \mathbf{0.189})}$ | $0.339_{(0.263,\ 0.440)}$ |
| | tsGT* | $\mathbf{0.085}_{(\mathbf{0.081},\ \mathbf{0.091})}$ | $\mathbf{0.155}_{(\mathbf{0.154},\ \mathbf{0.156})}$ | $0.178_{(0.158,\ 0.199)}$ | $0.298_{(0.219,\ 0.401)}$ |
| RMSE | t-student* | $0.216_{(0.213,\ 0.219)}$ | $0.486_{(0.484,\ 0.488)}$ | $0.513_{(0.448,\ 0.587)}$ | $0.543_{(0.389,\ 0.768)}$ |
| | Laplace | $0.218_{(0.215,\ 0.221)}$ | $0.488_{(0.486,\ 0.490)}$ | $0.540_{(0.455,\ 0.642)}$ | $0.546_{(0.393,\ 0.769)}$ |
| | Gaussian | $0.214_{(0.211,\ 0.217)}$ | $0.482_{(0.480,\ 0.484)}$ | $0.513_{(0.438,\ 0.603)}$ | $0.556_{(0.402,\ 0.776)}$ |
| | IQN* | $0.218_{(0.215,\ 0.221)}$ | $0.612_{(0.609,\ 0.615)}$ | $0.506_{(0.438,\ 0.585)}$ | $0.628_{(0.450,\ 0.858)}$ |
| | Informer | $0.403_{(0.398,\ 0.407)}$ | $0.343_{(0.340,\ 0.346)}$ | $0.347_{(0.343,\ 0.350)}$ | $0.394_{(0.383,\ 0.410)}$ |
| | ETSformer | $0.351_{(0.347,\ 0.356)}$ | $0.326_{(0.323,\ 0.329)}$ | $0.326_{(0.323,\ 0.330)}$ | $0.356_{(0.345,\ 0.372)}$ |
| | FEDformer | $0.324_{(0.321,\ 0.328)}$ | $0.293_{(0.290,\ 0.296)}$ | $0.283_{(0.279,\ 0.286)}$ | $\mathbf{0.334}_{(\mathbf{0.323},\ \mathbf{0.349})}$ |
| | Autoformer | $0.314_{(0.310,\ 0.318)}$ | $0.295_{(0.292,\ 0.298)}$ | $0.292_{(0.289,\ 0.296)}$ | $\mathbf{0.329}_{(\mathbf{0.318},\ \mathbf{0.344})}$ |
| | DLinear | $0.124_{(0.121,\ 0.128)}$ | $0.352_{(0.351,\ 0.354)}$ | $\mathbf{0.209}_{(\mathbf{0.187},\ \mathbf{0.233})}$ | $0.501_{(0.347,\ 0.720)}$ |
| | PatchTST | $\mathbf{0.117}_{(\mathbf{0.115},\ \mathbf{0.121})}$ | $0.320_{(0.318,\ 0.321)}$ | $\mathbf{0.208}_{(\mathbf{0.186},\ \mathbf{0.232})}$ | $0.464_{(0.310,\ 0.687)}$ |
| | TSMixer | $0.124_{(0.119,\ 0.131)}$ | $0.317_{(0.315,\ 0.318)}$ | $\mathbf{0.199}_{(\mathbf{0.178},\ \mathbf{0.222})}$ | $0.489_{(0.336,\ 0.713)}$ |
| | tsGT* | $\mathbf{0.114}_{(\mathbf{0.109},\ \mathbf{0.121})}$ | $\mathbf{0.285}_{(\mathbf{0.283},\ \mathbf{0.287})}$ | $\mathbf{0.220}_{(\mathbf{0.195},\ \mathbf{0.246})}$ | $0.455_{(0.297,\ 0.678)}$ |

* Stochastic model

Table 14: Performance of the models in terms of quantile metrics QL and CRPS (lower values are better). Each entry is an average with 90% bootstrap confidence intervals (in brackets) computed over $W = 100$ training windows and all time series in each dataset.

| metric | model | electricity | traffic | ETTm2 | weather |
|--------|-------|-------------|---------|-------|---------|
| CRPS | t-student | 0.14 (0.139, 0.145) | 0.28 (0.274, 0.277) | 0.46 (0.398, 0.529) | 0.37 (0.296, 0.47) |
| | Laplace | 0.14 (0.14, 0.146) | 0.27 (0.274, 0.276) | 0.48 (0.405, 0.56) | 0.37 (0.297, 0.471) |
| | Gaussian | 0.14 (0.136, 0.141) | 0.27 (0.266, 0.268) | 0.45 (0.385, 0.523) | 0.39 (0.317, 0.496) |
| | IQN | 0.14 (0.138, 0.144) | 0.32 (0.322, 0.325) | 0.44 (0.373, 0.511) | 0.38 (0.3, 0.49) |
| | tsGT | **0.06** (0.061, 0.066) | **0.12** (0.116, 0.118) | **0.14** (0.127, 0.161) | **0.24** (0.168, 0.342) |
| $QL_{50\%}$ | t-student | 0.17 (0.163, 0.169) | 0.31 (0.313, 0.315) | 0.47 (0.403, 0.534) | 0.39 (0.317, 0.492) |
| | Laplace | 0.17 (0.164, 0.17) | 0.31 (0.313, 0.315) | 0.49 (0.411, 0.586) | 0.39 (0.32, 0.495) |
| | Gaussian | 0.16 (0.162, 0.168) | 0.32 (0.314, 0.316) | 0.47 (0.396, 0.551) | 0.4 (0.328, 0.503) |
| | IQN | 0.17 (0.164, 0.17) | 0.4 (0.397, 0.4) | 0.45 (0.39, 0.531) | 0.41 (0.327, 0.518) |
| | tsGT | **0.08** (0.078, 0.085) | **0.15** (0.146, 0.147) | **0.17** (0.155, 0.194) | **0.27** (0.196, 0.373) |
| $QL_{75\%}$ | t-student | 0.15 (0.151, 0.159) | 0.31 (0.308, 0.311) | 0.44 (0.386, 0.505) | 0.47 (0.36, 0.621) |
| | Laplace | 0.16 (0.153, 0.161) | 0.31 (0.31, 0.313) | 0.46 (0.392, 0.536) | 0.48 (0.367, 0.629) |
| | Gaussian | 0.15 (0.147, 0.155) | 0.3 (0.298, 0.301) | 0.44 (0.379, 0.506) | 0.51 (0.392, 0.66) |
| | IQN | 0.16 (0.152, 0.16) | 0.39 (0.387, 0.39) | 0.41 (0.355, 0.475) | 0.47 (0.353, 0.618) |
| | tsGT | **0.07** (0.067, 0.078) | **0.13** (0.133, 0.135) | **0.16** (0.14, 0.187) | **0.33** (0.218, 0.482) |
| $QL_{95\%}$ | t-student | 0.1 (0.099, 0.109) | 0.24 (0.234, 0.237) | 0.4 (0.335, 0.467) | 0.5 (0.356, 0.693) |
| | Laplace | 0.1 (0.096, 0.106) | 0.23 (0.226, 0.229) | 0.4 (0.337, 0.47) | 0.49 (0.347, 0.685) |
| | Gaussian | 0.1 (0.092, 0.101) | 0.21 (0.204, 0.207) | 0.33 (0.283, 0.389) | 0.46 (0.321, 0.648) |
| | IQN | 0.1 (0.092, 0.102) | 0.22 (0.216, 0.219) | 0.34 (0.285, 0.407) | 0.45 (0.311, 0.64) |
| | tsGT | **0.03** (0.029, 0.032) | **0.07** (0.067, 0.068) | **0.09** (0.073, 0.111) | **0.31** (0.168, 0.498) |

