# OpenReview forum: "tsGT: Time Series Generative Transformer"
_TMLR — Rejected by TMLR_

### Review · Reviewer_fYHL · 2024-07-29

**Summary Of Contributions:**

This paper proposes a new time series forecasting model based on a vanilla decoder-only Transformer and a specially-designed discretization for real-world values, named tsGT. tsGT performs well in both deterministic and stochastic settings. The code is also public.

**Audience:**

Yes

**Broader Impact Concerns:**

They have discussed the Broader Impact.

**Claims And Evidence:**

Yes

**Requested Changes:**

The authors need to clarify the experiment implementations and add more baselines.

**Strengths And Weaknesses:**

Strengths

-	This paper is well-written and clear.
-	The proposed method is reasonable and novel, especially the special discretization method.
-	Extensive experiments and diverse metrics are included.
-	The experiments of Quick Draw datasets are interesting.


Weaknesses

1.	Some advanced baselines are missing.

Several latest time-series forecasting models should be considered.

[1] TimeMixer: Decomposable Multiscale Mixing for Time Series Forecasting, ICLR 2024

[2] TimesNet: Temporal 2D-Variation Modeling for General Time Series Analysis, ICLR 2023

2.	About implementations

-	Does tsGT treat multivariate time series in a channel-independent way?

-	Since the discretization will extend the input length with p times, how about the model efficiency, such as running time and GPU memory?

-	Are the experiments on Quick Draw datasets implemented as an imputation task? How to handle the varied length of input data?

---

### Review · Reviewer_8qK8 · 2024-08-05

**Summary Of Contributions:**

The paper proposes the decoder only transformer like architecture for time series prediction. It relies on discretization for input and output representation, and chain rule to recover the joint probability via composing the conditional distribution - to make the model probabilistic. It is evaluated on common benchmark datasets against the several baselines, in a rolling window evaluation protocol. Empirical results suggest better performance in terms of reduced error metrics as RMSE and MAE, as well as in quantile metrics as QL and CRPS.

**Audience:**

Yes

**Claims And Evidence:**

No

**Requested Changes:**

Questions:
- According to the text, the values are normalized to closed interval [0, 1] which includes both 0.000 and 1.000 (if we take p=3 as in the text). Given that only fixed number of decimal positions are taken for the representation of the value, how are minimal and maximal value distinguished?
- Why were not newer transformer architectures taken for a baselines, like for example: LagLama, Timer or TimesFM?

Suggestions:
- As applications and objectives may vary, I feel that readers would benefit from having insight into empirical evaluation against the baselines both in "fixed" (more commonly used setting) and "rolling-window" settings
- To test the hypothesis of "three aspects", an ablation study could be performed to see how much each of the aspects is contributing to the increase of the performance

**Strengths And Weaknesses:**

Strengths:
- In a moving window evaluation setting, it was shown to outperform the selected baselines
- It was shown empirically that the proposed model is not invariant to the order of inputs
- It is apparent that a lot of the computation resources were used for this empirical analysis

Weaknesses;
- Claims and hypotheses does not seem well supported. For example, it is suggested that three aspect can explain observed strong empirical results: stochastic nature of tsGT; transformer's dense learning; and nonparametric distributions capturing stochastic properties of time-series. However, there was no theoretical or empirical treatment applied to test those hypotheses.
- The baseline models are not the most recent ones, especially the transformer based ones which are mostly form 2022
- Even though baseline methods are developed and optimized for the single-fixed model setting, in this paper they are subjected only to a rolling window evaluation protocol

---

### Review · Reviewer_2pHH · 2024-08-06

**Summary Of Contributions:**

This work proposed a novel decoder-only transformer model tsGT for time series prediction, which discretizes the real-valued data into a stream digits for modeling.

**Audience:**

Yes

**Claims And Evidence:**

Yes

**Requested Changes:**

It is interesting to conduct multitask learning for the time series to see if different tasks can benefits with each other.

**Strengths And Weaknesses:**

Strengths:
(1) It is interesting to model time series by converting the real-value into stream digits for modeling.
(2) It compares multiple models with a variety of metrics

Weaknesses:
(1) The novelty for tsGT is limited since the time series tokenization method was proposed in the literature and the decoder model does not have any novel archetecture.

---

### Review · Reviewer_JxhS · 2024-08-06

**Summary Of Contributions:**

In this paper, the authors have proposed tsGT, a transformer-based stochastic time series model. The authors have also evaluated tsGT and compared it with the state-of-the-art algorithms on several benchmark datasets, such as electricity, traffic, ETTm2, and weather.

**Audience:**

Yes

**Broader Impact Concerns:**

None.

**Claims And Evidence:**

Yes

**Requested Changes:**

- Please clarify the novelty of the proposed model.
- Might the authors add more experiments to demonstrate how the performance of tsGT varies with the choice of the normalization constant, the precision, and $\beta$ in the loss function? I think it is important to understand if the performance of tsGT is sensitive to the choice of such parameters.
- The authors have shown that tsGT has outperformed most state-of-the-art benchmarks, which is great. However, I think a more fair comparison should also take the model size into consideration. Specifically, might the authors add scatter plots to show the model size-performance tradeoff? The x-axis of such plots can be the model size, and the y-axis can be the performance (e.g. RMSE), and we can have four such plots, one for each dataset. The goal of this plot is to justify the performance improvement is (mainly) due to the better architecture (transformer), rather than larger models.

**Strengths And Weaknesses:**

**Strengths:**

- Overall, the paper is well written.
- The authors have done a good job of literature review.
- The experiment results in this paper seem to be solid. The authors have compared the proposed tsGT with extensive state-of-the-art benchmarks, on four datasets, and under several different metrics (see Table 1-3).

**Weaknesses:**
- My main concern is that the proposed model, tsGT, seems to lack novelty and originality. It seems to be a straightforward application of the decoder-only transformer. The main novelties seem to be the input and output discretization, and the loss function (in particular its dependence on $\beta$).
- For the data processing, we need to choose a normalization constant and a fixed precision. It is not completely clear how to choose them. Appendix A.2 has briefly discussed how to choose the normalization constant, but how to choose $r$ in this approach? Also, how should we choose the fixed precision? Intuitively, if the precision is too high, we will end up with many tokens for one number and the transformer needs to capture longer-range dependencies; on the other hand, if the precision is too low, it will incur some performance loss due to the missing information. How should we choose the precision?
- Similarly, it is not clear to me how to choose $\beta$ in the loss function.

---

### Comment · Action_Editor_1n1D · 2024-09-02
**Rebuttal?**

Dear Authors,

Would you like to enter a rebuttal?

AE.

---

### Decision · Action_Editor_1n1D · 2024-09-17

**Recommendation:** Reject

**Comment:**

The decision to recommend rejection is based on the consensus among three out of four reviewers who lean towards rejection, citing insufficient evidence to support the paper's claims. While acknowledging the paper's strengths, including well-conducted experiments on multiple datasets, reviewers raise significant concerns. These include the lack of comparison to the most recent baseline models, insufficient analysis of the model's performance drivers, inadequate exploration of hyperparameter sensitivity. Reviewers suggested addressing these issues through additional experiments, ablation studies, and more comprehensive comparisons. The authors' lack of rebuttal to address these concerns further weakens the case for acceptance.

**Audience:**

The paper can be of interest to researchers working on time series models.

**Claims And Evidence:**

Based on the reviewer feedback, the claims made in the submission are not fully supported by convincing evidence. While the paper presents promising results for the tsGT model on several benchmark datasets, there are gaps in the evidence provided. The main concerns include: (1) lack of comparison to the most recent state-of-the-art models, particularly transformer-based approaches from 2023-2024, which makes it difficult to assess the true performance gains, (2) insufficient analysis to support claims about why the model performs well, with no ablation studies to isolate the contributions of different model aspects, (3) inadequate exploration of hyperparameter sensitivity and optimal choices, and (4) absence of model size vs. performance trade-off analysis against baselines. These issues, combined with the lack of a rebuttal addressing reviewer concerns, undermine the strength of the evidence supporting the paper's claims.